# Optimization of Torque-Control Model for Quasi-Direct-Drive Knee Exoskeleton Robots Based on Regression Forecasting

**DOI:** 10.3390/s24051505

**Published:** 2024-02-26

**Authors:** Yuxuan Xia, Wei Wei, Xichuan Lin, Jiaqian Li

**Affiliations:** 1School of Optoelectronic Science and Engineering, Soochow University, Suzhou 215031, China; 20214239034@stu.suda.edu.cn; 2School of Electronic & Information Engineering, Suzhou University of Science and Technology, Suzhou 215009, China; 3MebotX Intelligent Technology (Suzhou) Co., Ltd., Suzhou 215131, China; linxichuan@live.com

**Keywords:** wearable sensors, human movement, torque control, knee exoskeleton, regression forecasting

## Abstract

The choice of torque curve in lower-limb enhanced exoskeleton robots is a key problem in the control of lower-limb exoskeleton robots. As a human–machine coupled system, mapping from sensor data to joint torque is complex and non-linear, making it difficult to accurately model using mathematical tools. In this research study, the knee torque data of an exoskeleton robot climbing up stairs were obtained using an optical motion-capture system and three-dimensional force-measuring tables, and the inertial measurement unit (IMU) data of the lower limbs of the exoskeleton robot were simultaneously collected. Nonlinear approximations can be learned using machine learning methods. In this research study, a multivariate network model combining CNN and LSTM was used for nonlinear regression forecasting, and a knee joint torque-control model was obtained. Due to delays in mechanical transmission, communication, and the bottom controller, the actual torque curve will lag behind the theoretical curve. In order to compensate for these delays, different time shifts of the torque curve were carried out in the model-training stage to produce different control models. The above model was applied to a lightweight knee exoskeleton robot. The performance of the exoskeleton robot was evaluated using surface electromyography (sEMG) experiments, and the effects of different time-shifting parameters on the performance were compared. During testing, the sEMG activity of the rectus femoris (RF) decreased by 20.87%, while the sEMG activity of the vastus medialis (VM) increased by 17.45%. The experimental results verify the effectiveness of this control model in assisting knee joints in climbing up stairs.

## 1. Introduction

In recent years, exoskeleton robots have shown great potential in enhancing the endurance of human movement [1,2]. Compliance, light weight, and the ability to assist multiple movements are key requirements for exoskeleton robots to be viable in everyday life [3]. Lower-limb exoskeletons that directly provide joint torque through actuators or passive components can be used to enhance human motion [4]. The assisting principle is to provide energy to the lower-limb joints in order to directly reduce the activation of the associated muscles and the resulting energy consumption. There have been many research studies conducted on hip or ankle exoskeleton robots that enhance walking ability. In contrast, there has been little research on knee exoskeleton robots, possibly because the positive work performed by the knee during horizontal walking is less than the positive work performed by the hip or ankle [5]. However, there are more advantageous solutions for moving on flat surfaces, such as wheeled robots. The advantages of exoskeleton robots are more reflected in their adaptability to complex road conditions, such as stair climbing and mountain climbing. In these environments, the help of the knee joint is critical.

In recent years, there have been many research studies on torque assistance for enhanced lower-limb exoskeleton robots. Jin et al. [6] used a proxy-based sliding mode controller to track the expected force, and the experimental results showed that the energy consumption was reduced by about 5.9% on average. Chen et al. [7] proposed the use of a Rayleigh oscillator to fit gait information, and this method could maintain relatively stable predictions under the condition of large changes in the gait period. Kim et al. [8] proposed a human-in-the-loop Bayesian optimization algorithm that combined two parametric sinusoids at the peak of assistance to apply an assist curve to aid in hip extension. Guo et al. [9] proposed a new type of assist function that can generate the desired assist function only by adjusting three parameters: the assist amplitude, the assist period, and the assist peak shift. Arefeen et al. [10] used a two-dimensional human exoskeleton model and a powered knee exoskeleton robot to predict the optimal lifting motion and assist torque. Gordon et al. [11] proposed a human-in-the-loop method that uses Bayesian optimization to drive the selection of control parameters between loop protocol cycles and optimizes the assist torque provided by the powered hip exoskeleton robot. Martinez et al. [12] used hip and knee angles to map joint motion to drive torque control of a lower-limb exoskeleton robot during swinging motion and allow the user to vary their step length. Lim et al. [13] used the hip angle to provide continuous sinusoidal torque assistance from a hip exoskeleton robot. Quintero et al. [14] used gait stages to map kinematic constraints in order to continuously provide torque assistance based on periodic waves for knee and ankle prosthetics. Thatte et al. [15] developed a continuous torque curve to assist knee and ankle prostheses by utilizing the joint angle, angular velocity, and feedforward torque. Huang et al. [16] conducted stiffness modeling for a lightweight quasi-direct-drive (QDD) knee exoskeleton robot and carried out continuous torque control based on stiffness, achieving a larger stiffness tracking bandwidth and a smaller torque tracking error, but their assist torque curve was only based on the proportional biological torque curve. Bryan et al. [17] studied the relationship between walking speed, exoskeleton power, and metabolic energy expenditure and concluded that the exoskeleton robot is most effective during moderate and fast walking and less effective during slow walking. Bryan et al. [18] also used a human-in-the-loop system to optimize the assist of a whole-lower-limb exoskeleton robot, and the consistent optimization of timing parameters showed that metabolic reduction was sensitive to torque timing. Miller et al. [19] used an ankle exoskeleton simulator to characterize the impact of peak assist torque on metabolic cost during running and found that the relationship between the metabolic rate and peak assist torque at low torque was almost linear, while the report decreased at high torque, showing a progressive exponential function.

However, most of the above research studies referred to the human joint torque curve to establish the assist function. On the basis of the assist function, researchers studied the relationship between each parameter of the function and the assisting effect and found the best parameter to establish a control model. On the one hand, the established assist function loses some biological characteristics relative to an actual human joint torque curve. The curve generated by the function does not conform to the characteristics of the human joint force and thus cannot achieve the best assisting effect. On the other hand, there is a difference between the ideal human joint torque curve and the actual output curve of the exoskeleton robot system to the joint. First, there is an inevitable gap in the structural connection of the exoskeleton robot system, resulting in a mechanical transmission delay. Second, the communication between the main control, sensing, and driving subsystems of the exoskeleton robot will cause delays. Finally, when torque control is carried out, traditional control methods such as PID are often used to track the torque in the bottom drive of the motor actuator, and there is a delay caused by increasing time.

With the development of modern science, in order to solve the problem of exoskeleton robots being difficult to model through mathematical tools, some researchers have applied neural networks to the control of exoskeleton robots. Ren et al. [20] proposed a neural network and LSTM machine learning model method to predict the actual motion trajectory of human lower limbs, which can accurately predict the gait trajectory with small root mean squared error (RMSE). Lee et al. [21] used a deep learning slope prediction model that was able to generalize between users and terrain. They collected training data and used CNN to predict the inclination angle of the knee exoskeleton robot in real time and actively adjust the peak assist amplitude. The average RMSE of the predicted slope of the model was 1.5. Wang et al. [22] proposed a genetic algorithm–back propagation (GA-BP) neural network to estimate the wearer’s motion intention through sEMG signals. Wu et al. [23] proposed a graph convolutional network model (GCNM) for gait phase classification of the lower-limb exoskeleton, which can solve the non-Euclidean domain gait phase classification problem based on the exoskeleton diagram mechanism. Kang et al. [24] used a CNN-based gait phase estimator that adapts to different motion mode settings to regulate the assist of the exoskeleton robot, and the RMSE estimated for gait stages was 5.04 ± 0.79%. Zhang et al. [25] proposed a comprehensive network model that combines a sparse autoencoder (SAE), bidirectional long short-term memory (BiLSTM), and deep neural network (DNN). The model can accurately identify the four stages in the gait cycle, and the accuracy and F-value are better than other algorithms. Xia et al. [26] used a model based on CNN-BiLSTM to classify seven gait phases of both legs through IMU data of lower limb and plantar pressure data, with a maximum accuracy of 95.09%. Chen et al. [27] proposed a lower-limb exoskeleton gait pattern-recognition method based on LSTM and CNN, which can recognize five common gait patterns with an average recognition accuracy of 97.78%. Wang et al. [28] proposed an integral subject-adaptive real-time locomotion mode recognition (LMR) method based on GA-CNN for a lower-limb exoskeleton system. They adopted the Bayesian optimization method to optimize the hyper-parameters, which could recognize twelve locomotion modes.

The construction of a neural network model often requires a large amount of data for model training, including the sensor data used as the input to the model and the corresponding output results. Most of the above research studies focused on the prediction of sensor data time or gait phase classification. Time series prediction can use the real data of the next moment as the result of network training, and gait phase classification can obtain the correct classification result through images or other sensors. However, the above research studies can not directly obtain the output torque of the motor actuator. After obtaining the sensor data at the next moment or the gait phase at the moment, the control model needs to be established to realize the assistance of the exoskeleton robot system. This undoubtedly increases the complexity and operating time of the entire control system.

Based on the above problems, the main work of this paper is as follows. Firstly, we measured the knee torque of five subjects when they were climbing the stairs through an optical motion-capture system and three-dimensional force-measuring tables. In this process, we simultaneously collected the IMU data of lower limbs applied to the control of the exoskeleton robot. Secondly, we design a regression forecasting network that combines CNN and LSTM. This network has the ability to extract data features and process time series well. The input of the network is the IMU data of lower limbs, and the output is the torque of the knee joint. In order to compensate for the delay caused by mechanical transmission, communication between subsystems, and motor control at the bottom level, the knee torque data were time-shifted to different degrees in the model-training stage, and multiple torque-control models were obtained. We propose a model optimization index to evaluate the assisting effect of the exoskeleton robot torque-control model using the degree of overlap between the ideal curve and the actual curve of the motor actuator. Finally, the above control model and optimization method were verified by sEMG experiments. The results of sEMG experiments show that the optimization index based on the coincidence degree of the ideal curve and actual curve can reflect the assisting effect of the exoskeleton robot. With the assistance of the above model, the sEMG activity of the RF decreased by 20.87%, while the sEMG activity of the VM increased by 17.45%.

## 2. Materials and Methods

### 2.1. Knee Exoskeleton Robot

The experimental platform used in this research study was based on the knee exoskeleton robot developed by MebotX Intelligent Technology (Suzhou) Co., Ltd., Suzhou, China. The exoskeleton robot was mainly used to walk up and down stairs. The main components of the exoskeleton robot included the waist pack module, the driving system, the binding system, the thigh frame, the calf frame, and the sensing system, as shown in Figure 1. The waist pack has the main control and power system attached to it. The binding system was used to connect each module and fix the robot to the human body. The driving system included Quasi-Direct-Drive (QDD) actuators for the left and right knee joints. QDD actuators connected the thigh frame to the calf frame. The sensing system consisted of two torque sensors and six IMUs. The torque sensors were, respectively, installed between the left and right knee actuators and the frames; the IMUs were, respectively, located in the left and right thighs, left and right calves, and left and right feet.

The knee exoskeleton robot demonstrated exceptional human–machine compatibility. First, the curved shape of the thigh and calf frames was derived from inverted leg models of various potential users. The main material of the frame was carbon fiber, guaranteeing both structural strength and lightweight properties with a mere 1.8 kg weight per leg. There were two kinds of binding materials with different stiffness on the inside of the frame to ensure effective force transfer and a comfortable fit with the human body. There was a flexible connection between the waist and the thigh, which would not interfere with the movement of the wearer. The knee of the robot had a flexion motion range of −5° to 160° and was suitable for activities that require a wide range of motion, such as walking up and down stairs, sitting, and squatting. In addition, different size adjustment structures were designed at the top of the thigh and the end of the calf, ensuring that the exoskeleton robot could adapt to various body types.

In this study, the knee exoskeleton robot was equipped with a QDD actuator in each active joint. The actuator was based on a lightweight motor and a planetary reducer with low-slotted torque and had a fully integrated drive system, as shown in Figure 2.

To meet the design and control requirements, we customized a QDD actuator that was lightweight (521 g) and compact (Φ89 × 50.25 mm height) and produced a peak torque of 24.8 Nm. The QDD system consisted of a high torque density DC brushless motor with a peak torque output capability of 2.48 Nm, a planetary reducer with a reduction ratio of 10:1, a 14-bit magnetic encoder (AS5048, Premstaetten, Austria), and a motor driver (controller STM32F405RGT6, ST, Paris, France). Unlike actuators that use high-ratio mechanisms, such as harmonic reducers and SEA actuators that use spring mechanisms, QDD actuators are based on low-ratio designs with no spring structure. In terms of mechatronics design, QDD actuators are simpler and more effective.

We implemented a low-level control loop in the motor microcontroller for position, speed, and current control. Real-time communication was carried out between microcontroller and computer through CAN bus protocol. Powered by a nominal voltage of 24 V, the actuator could reach a nominal speed of 310 r/min. In addition, due to the low reduction ratio gear drive of the QDD actuator, the actuator had low output inertia (32.2 kg·cm^2^), which was essential to achieve low impedance, thereby minimizing the resistance to natural movement of the human body.

### 2.2. Experimental Data Acquisition

In this study, the torque of knee joint during human movement was obtained using optical motion-capture system. Vicon is a optical motion-capture system developed by Vicon Motion Systems Ltd., Oxford, UK. The device can establish a complete lower-limb dynamic model and deduce joint torque during human movement, so it is often used in ergonomic and biomechanical analysis. Vicon equipment included MX cameras, MX-Ultranet Unit, computer, Vicon Datastation ADC Patch Panel, Kistler three-dimensional force-measuring tables, MX special cable, calibration kit (T-shaped calibration rack), and standard accessories (reflective markers, double-sided tape, etc.). There were eight MX cameras, which were evenly installed around the laboratory to collect optical data on human movement. The Vicon Datastation ADC Patch Panel was used to synchronously collect data from third-party test systems. The Kistler three-dimensional force-measuring tables were used to collect the ground reaction force during human movement.

The Vicon system was calibrated, and the subject’s leg length, knee width, and ankle width were measured before data collection began. This study involved the assistance of the exoskeleton robot to the human body when climbing up stairs, so we built two steps through planks on the force-measuring tables; the height of each step was 20 cm. The model used in this study was “Vicon PlugInGait Lowerbody Ai”, which required 16 reflective markers to be attached to the lower extremities, 1 on each side of the posterior superior iliac, anterior superior iliac, thigh, knee, tibia, ankle, toe, and heel. The schematic diagram of the reflective markers is shown in Figure 3.

To facilitate the adhesion of the reflective markers, the frames, QDD actuators, and most of the bindings of the exoskeleton robot should be removed during data collection, and the main control and IMUs should be retained. The input of the torque-control model in this study was the motion data of the thigh, calf, and foot, including triaxial acceleration, triaxial angular velocity, pitch angle, and roll angle. These data were collected using IMUs (YIS106-V, Wuhan Yuansheng Innovation Technology Co., Ltd., Wuhan, China). The sensor had a pitch angle measuring range of ±90° and a roll angle measuring range of ±180°, and the dynamic accuracy could reach 0.5°. The sensor had an angular velocity range of ±2000°, an acceleration range of ±16 g, and a maximum acquisition frequency of 200 Hz. Figure 3 shows the installation position of the sensor. The output of the torque-control model in this study was the knee torque data, which were obtained through the Vicon system. Vicon Datastation ADC Patch Panel can collect analog voltage and display and store it synchronously with the data of the Vicon system. Therefore, this study used the CAN-to-analog module to convert the IMU data of the exoskeleton robot into 0–5 V analog voltage to realize time synchronization with the knee torque data. The block diagram of the data-acquisition system is shown in Figure 4.

To improve the generalization performance of the model, we collected experimental data from people with different heights and weights. Four healthy male subjects and one healthy female subject, all between the ages of 25 and 30, participated in the study. Table 1 provides the details of the subjects. All participants participated voluntarily and could stop at any time. The experiment was reviewed and approved by the Ethics Committee of Soochow University, numbered SUDA20221228H08.

Due to the small number of steps of force-measuring stairs, the subject needed to climb up the force-measuring stairs repeatedly during data collection. Each subject was required to climb up stairs ten times. Each time the subjects climbed up stairs, they stepped with their left foot first five times and then their right foot first five times. Only the data of the subject on the force-measuring tables were retained. We retained data through the resultant force of the three-dimensional force tables. When the volunteer’s sole contacted the first force table, the resultant force of the force table started to change from 0. We marked the first non-zero resultant force obtained from the first force table as the starting point of the data. When the volunteer’s sole detached from the contact with the second force table, the resultant force of the force table changed to 0. We marked the last non-zero resultant force obtained from the second force table as the end point of the data. A total of 50 sets of data were collected by the above 5 subjects.

### 2.3. CNN-LSTM Regression Forecasting Model

CNN has a strong ability for data feature extraction, so it can effectively extract the coupling features between the acceleration, angular velocity, and Euler angle of the lower limbs and establish the corresponding feature vector. CNN includes a convolutional layer for feature extraction, a pooling layer for feature selection and filtering, and a fully connected layer for feature conversion to vector output. The main structure of CNN is shown in Figure 5.

The core of CNN is the convolution layer, and each convolution layer is composed of a different convolution kernel. The convolution kernel will perform operations on the input data and map it to features. Features are dimensionalized in the pooling layer. According to the dimension and size of the collected data, this paper chose CNN’s representative LeNet model. The typical structure of the model has three convolution layers, two pooling layers, and two fully connected layers, which have the characteristics of simple structure and strong application. After feature extraction and flattening by LeNet model, the motion data of lower limbs were introduced into LSTM model for knee torque prediction.

LSTM can extract feature relationships of long and short time series and performs well in processing time series. This is because LSTM introduces a “gate mechanism” to solve the long-term memory of time series information. By controlling the input gate, the forget gate, and the output gate, the LSTM adds or deletes current and past time state information to the storage unit. The LSTM structure diagram is shown in Figure 6.

In the LSTM, the status-update process of the storage unit is as follows.

The forget gate deletes information from the storage unit based on the input of the current time t and the output of the previous time t−1. The expression is
(1)$$f_{t}=\sigma \left(w_{fh}\times h_{t-1}+w_{fx}\times x_{t}+b_{f}\right)$$

In Formula (1), $f_{t}$ is the state of the forgetting gate, $w_{fh}$ and $w_{fx}$ are the weight matrix of the forgetting gate, $b_{f}$ is the offset of the forgetting gate, and σ is the activation function.

The input gate determines the information stored in the storage unit. The expression is
(2)$$i_{t}=\sigma \left(w_{ih}\times h_{t-1}+w_{ix}\times x_{t}+b_{i}\right)$$
(3)$${\tilde{c}}_{t}=tanh\left(w_{\tilde{c}h}\times h_{t-1}+w_{\tilde{c}h}\times x_{t}+b_{\tilde{c}}\right)$$

In Formulas (2) and (3), $i_{t}$ is the state of the input gate, ${\tilde{c}}_{t}$ is the state of the candidate element, the weight matrix of the input gate is $w_{ih}$, $w_{ix}$ and $w_{\tilde{c}h}$, the offset of the input gate is $b_{i}$ and $b_{\tilde{c}}$, and tanh and σ are the activation function.

Update the unit status $C_{t}$ according to the state of the forget gate and the input gate; the expression is
(4)$$C_{t}=f_{t}\times C_{t-1}+i_{t}\times {\tilde{c}}_{t}$$

In Formula (4), $C_{t-1}$ is the state of the unit at time t−1.

In the current state of the unit, update the state $O_{t}$ of the output gate; the expression is
(5)$$O_{t}=\sigma \left(w_{oh}\times h_{t-1}+w_{ox}\times x_{t}+b_{o}\right)$$

In Formula (5), $w_{oh}$ and $w_{ox}$ are the weight matrix of the output gate; $b_{o}$ is the offset of the output gate.

The final output of the output gate is
(6)$$h_{t}=O_{t}\times tanh\left(C_{t}\right)$$

In Formula (6), $h_{t}$ is the output of time t, which is the output torque of the knee joint.

In this paper, the CNN-LSTM combined model was used to predict the knee output torque. Compared with single CNN and single LSTM, this model had both strong ability to extract data features and process time series and was more suitable for knee joint output torque-prediction scenarios. In the combined model, the CNN network was used to process the historical data of the IMUs, and the output feature quantity was the coupling characteristic between motion data. The feature quantity extracted by CNN was converted into the time series and input into LSTM for regression prediction of knee output torque. The structure of the combined CNN-LSTM model is shown in Figure 7.

IMU has the advantages of light weight, high reliability, and no environmental restrictions in the control of exoskeleton robots. Some researchers have worked on the design of exoskeleton robot torque controllers using IMU data as input. Long et al. [29] proposed an online gait-generation method based on gait step calculation and direct measurement using IMU sensors to provide continuous assistance for walking with a knee exoskeleton robot. Yu et al. [30] proposed a biomechanical model-based control method that generated an assistive torque profile for versatile control of both squat and stoop lifting assistance. The method used IMUs to detect the lifting posture and generated an assistive profile proportional to the human joint torque generated by the model. Liu et al. [31] proposed a muscle force estimation method based on the neural network model, which used the kinematic data collected by IMUs as input, and the muscle force of gluteus maximus, rectus femoris, gastrocnemius, and soleus as output. Muscles generate forces by contracting and relaxing, which are applied to the joints, resulting in joint movement. The above studies reflected the correlation between the joint torque of the lower limbs during human movement and the kinematic data of the lower limbs, which can be collected by IMU. Therefore, this study used the kinematic data collected by IMUs as model input to predict the torque of the knee joint.

The input matrix X of the model was composed of the triaxial acceleration, triaxial angular velocity, pitch angle, and roll angle of each leg segment, and the number of sensor data was 48 per iteration. In the training and prediction of the model, the sliding window slid on the matrix to obtain the corresponding value into the model and carry out feature extraction and time series prediction. Considering the long and short memory functions of LSTM, the size of the sliding window was set to 48, and the step size was set to 1. The data of the left thigh, left calf, left foot, right thigh, right calf, and right foot were Entered at the same time; that was, a matrix of 48 × 48 was obtained. The model input matrix was expressed as
(7)$$X=\left[\begin{array}{c} \begin{array}{c} X_{lt}\\ X_{lc} \end{array}\\ \begin{array}{c} X_{lf}\\ X_{rt} \end{array}\\ \begin{array}{c} X_{rc}\\ X_{rf} \end{array} \end{array}\right]=\left[\begin{array}{ccc} \begin{array}{c} x_{lt}\left(t-47\right)\\ x_{lc}\left(t-47\right)\\ x_{lf}\left(t-47\right) \end{array} & \begin{array}{c} x_{lt}\left(t-46\right)\\ x_{lc}\left(t-46\right)\\ x_{lf}\left(t-46\right) \end{array} & \begin{array}{cc} \begin{array}{c} \cdots \\ \cdots \\ \cdots \end{array} & \begin{array}{c} x_{lt}\left(t\right)\\ x_{lc}\left(t\right)\\ x_{lf}\left(t\right) \end{array} \end{array}\\ x_{rt}\left(t-47\right) & x_{rt}\left(t-46\right) & \begin{array}{cc} \cdots & x_{rt} \end{array}\left(t\right)\\ \begin{array}{c} x_{rc}\left(t-47\right)\\ x_{rf}\left(t-47\right) \end{array} & \begin{array}{c} x_{rc}\left(t-46\right)\\ x_{rf}\left(t-46\right) \end{array} & \begin{array}{cc} \begin{array}{c} \cdots \\ \cdots \end{array} & \begin{array}{c} x_{rc}\left(t\right)\\ x_{rf}\left(t\right) \end{array} \end{array} \end{array}\right]$$

In Formula (7), $X_{lt}$, $X_{lc}$, $X_{lf}$, $X_{rt},X_{rc}$, and $X_{rf}$ represented the feature matrix composed of the motion data of the left thigh, left calf, left foot, right thigh, right calf, and right foot. The size of each matrix was 6 × 48, where 6 was the number of data for each sensor and 48 was the length of the sliding window. Taking the prediction of the output torque of the left and right knee joints at time t as an example, the lower limb motion data were first input into the convolution layer, and then the convolution array was pooled in the maximum pooling layer to reduce the matrix size. After multiple convolutions and pooling of the input data, a 120 × 5 × 5 feature matrix was formed, which was flattened to form a 1 × 3000 time series. Then, the time series was input into the LSTM network to predict the knee torque. Finally, the predicted output torque of the left and right knee joints at time t was output in the fully connected layer.

*RMSE* is a commonly used measure of the difference between the predicted value of a model and the actual observed value, which is used to evaluate the degree of fitting of a model to a given data. *RMSE* is obtained by calculating the mean of the square of the difference between the predicted value and the actual observed value and taking its square root. The value of RMSE is the same as the unit of the original observation. It can measure the average size of the prediction error of the model. A smaller *RMSE* indicates that the difference between the predicted value of the model and the actual observed value is less, and it means the model fits better. The advantage of RMSE is that there is a larger penalty for large error values because it squares the difference. This prevents large error values from affecting the fitting too much. In the process of moment control of exoskeleton robot, the large difference value will cause the robot to not assist the human body and hinder human movement. Therefore, *RMSE* was used as an evaluation index in this study. On the one hand, we used *RMSE* to evaluate the difference between the ideal torque data obtained using the motion-capture system and the three-dimensional force-measuring tables and the actual torque data output by the QDD actuator. On the other hand, we used RMSE to evaluate the difference between the regression forecasting model and the actual value. The RMSE formula is as follows:(8)$$RMSE=\sqrt{\frac{1}{n}\sum_{i=1}^{n}{\left(y_{i}-{\hat{y}}_{i}\right)}^{2}}$$

In Formula (8), RMSE is root mean square error, $y_{i}$ and ${\hat{y}}_{i}$ are the actual value and the predicted value, and n is the total number of the data.

### 2.4. Surface Electromyography Experiment

The effectiveness of the proposed model and optimization method was verified by sEMG experiments. SEMG signals were captured using the biological signals generated by the muscles during biological activities through the electrode guidance, and the captured signals were amplified and recorded to obtain ordered one-dimensional time series signals. SEMG signal acquisition has the characteristics of low risk, multi-target selection, and non-damage [32]. Several bipolar Ag/AgCl (Ambu A/S, Ballerup, Denmark) surface electrodes were installed at the RF and the VM after rubbing with alcohol at the test point of the subject’s back. The test point was based on Rogers et al. [33]. The locations are shown in Figure 8. The five subjects in Table 1 were still invited to participate in the sEMG experiment.

To obtain the real verification effect, the sEMG experiments were also carried out in a real environment. Subjects were asked to climb 12 stairs in a row each time, and video recordings of the experiment were created for data processing. We segmented the sEMG data of each leg, taking the time when the sole left the ground as the segmentation point, and each group of segmented data contained a set of support phase and swing phase. In order to ensure the consistency of climbing speed, the stepping rhythm of the subjects was controlled using prompt tone. To verify the effectiveness of the proposed regression forecasting network in exoskeleton torque control and the influence of differences in time shift parameter $t_{s}$ on the assisting effect, each subject conducted experiments with NO-EXO, EXO-OFF, $t_{s}=0$, $t_{s}=20\;ms$, and $t_{s}=50\;ms$. According to the analysis below, $t_{s}=50\;ms$ is the best time-shift parameter based on RMSE optimization. To reduce measurement errors, sEMG experiments were performed three times for each participant in each assisting condition. During signal collection, if the median frequency of the sEMG shifts to the left, indicating that the wearer has muscle fatigue, the sEMG experiment should be suspended for at least 2 h to ensure that the fatigue of the muscle returns to normal. During the whole experiment, a total of 450 groups of slicing data were collected from 5 subjects.

## 3. Results

The knee torque curve collected using the motion-capture system and the three-dimensional force-measuring tables was regarded as the ideal curve, and the motor actuator was controlled to track the curve. The actual torque exerted using the motor actuator on the human joint was measured using the torque sensors installed between the output position of the motor actuators and the frames of the exoskeleton robot. One set of curves is shown in Figure 9. The QDD actuator has the advantages of light weight, high dynamic response, and good reverse drive characteristics, but its output torque is relatively small. Therefore, it could not provide 100% torque assistance to the knee joint during the support phase, which resulted in a large gap between the actual curve and the ideal curve of the support leg in Figure 9.

We controlled the motor actuator to output each set of ideal torque curves of all subjects. According to Formula (8), the RMSE of all groups of the ideal torque data and the actual torque data was calculated, and then the average value was taken. We time-shifted the ideal curve with time $t_{s}$ and then calculated the RMSE for differences in the time-shift parameter $t_{s}$. The change curve of the weighted average accuracy with time shift parameters is shown in Figure 10.

It can be seen in the data in Figure 10 that when $t_{s}=50\;ms$, the mean of the RMSE is the smallest, and the degree of coincidence between the ideal curve and actual curve is the highest. We used $t_{s}=0$, $t_{s}=20\;ms$, and $t_{s}=50\;ms$ and then conducted regression forecasting training using the network in Section 2.3. The prediction results when $t_{s}=0$ are shown in Figure 11, and the RMSE under each parameter is shown in Figure 12.

SEMG signals reflected electromyographic activity and muscle output force. In the unfatigued state, after the original sEMG signal was processed by rectification, the 400 Hz low-pass filter, and root mean square (RMS) procedure, the greater amplitude of the sEMG signal, the greater force applied by the muscle. Time domain analysis and frequency domain analysis are commonly used in sEMG signal analysis. The analyzing method in time domain analysis included the root mean square value and average amplitude (MA), which could be used to judge the extent of muscle output force under normal conditions. The frequency domain analysis processing method includes the median frequency (MF), which indicated muscle fatigue. Figure 13 shows the measuring results of mean RMS values of the sEMG signal amplitude under various assisting situations. The red area is the 95% confidence interval for the amplitude of the sEMG signal under different positions and assisting situations. Each step of the climb lasted 0.75 s, and a gait cycle took 1.5 s. Table 2 shows the max RMS value under different assisting situations. The results of the sEMG experiments show that the optimization index based on the coincidence degree of ideal curve and actual curve can reflect the assisting effect of the exoskeleton robot. With the assistance of the above model, the sEMG activity of the RF decreased by 20.87%, while the sEMG activity of the VM increased by 17.45%.

## 4. Discussion

Through the analysis of the torque curves of the knee joints when the human body climbs the steps, it can be seen that when the heel touches the ground, the knee joint extends and drives the whole body to move forward and upward, so the knee joint of the supporting leg needs to output a large torque to promote such movement. When the knee joint of the other leg bends and moves to the next step during the swinging process, the amplitude of the required output torque is smaller. Through comparative analysis of the knee torque curve collected using the motion-capture system and the three-dimensional force-measuring tables and the motor actuator tracking torque curve, it can be found that the actual output torque curve of the motor actuator always lags behind. This lag means that there is a delay in the exoskeleton robot system, which may be caused by mechanical transmission, communication between subsystems, motor control at the bottom level, data filtering, etc. This also shows that this research study is meaningful. Considering the weight of the motor actuator and the resistance due to the large deceleration ratio, the maximum torque that the QDD actuator used in this study can output is 24.8 Nm. Therefore, when tracking the torque curve of the knee joint of the supporting leg, the peak of the ideal curve cannot be reached.

In this study, the RMSE was used to measure the difference between the knee torque curve collected using the motion-capture system and the three-dimensional force-measuring tables and the motor actuator tracking torque curve. It can be seen from the curve in Figure 10 that when the time-shift parameter $t_{s}=0$, there are differences between the two curves due to the above-mentioned lag, and the torque that the motor actuator can output cannot reach the peak torque of the knee joint of the supporting leg. When $t_{s}<0$, the lag increases, and the RMSE increases accordingly. When $t_{s}>0$, the lag is gradually compensated and the RMSE decreases. When $t_{s}=50\;ms$, the RMSE is reduced to the minimum, and the minimum RMSE is $4.64\times 10^{-3}\mathrm{Nm}/\mathrm{kg}$. When $t_{s}$ continues to increase, the actual torque curve of the motor output is ahead of the ideal torque curve, and the RMSE increases.

To verify the effectiveness of the model and parameter-optimization methods in this research study, three sets of prediction models were established, corresponding to $t_{s}=0$, $t_{s}=20\;ms$, and $t_{s}=50\;ms,\;respectively$. Based on the same data set, the three models have different degrees of time shift on the output data. As can be seen in Figure 11, the model we used can regression-forecast the knee torque data conforming to the human movement characteristics through the IMU data of each leg segment. In both the input IMU data and the output torque data, the variance of the supporting leg data is higher than that of the supporting leg. Therefore, the tracking effect of the regression-forecasting model on the swinging leg is better than that of the supporting leg. It can also be seen in Figure 12 that the RMSE of the supporting leg is higher than that of the swinging leg under different time-shift parameters. It can also be seen in Figure 12 that the RMSE increases as a whole with the increase in the time-shift parameter $t_{s}$, because when $t_{s}>0$, the regression-forecasting model predicts the knee torque data at the future time, which will produce some errors.

By observing the average amplitude of the RF and the VM sEMG signals when climbing stairs under the NO-EXO state (Figure 13), it can be seen that when the knee joint is bent and moves to the next step during the swinging process, the amplitude of the output torque is smaller. When the heel lands on the ground, the knee joint stretches to drive the whole body to move forward and upward, so the knee joint of the supporting leg needs to output a large torque, and the sEMG signal spikes significantly at this time.

According to the curve in Figure 13 and the data in Table 2, it can be seen that in the EXO-OFF state, the robot’s dead weight and the resistance of the motor actuator will increase the sEMG amplitude, and the VM increases by 8.57% compared with the NO-EXO state. The two muscles increase by 3.81% as a whole. The small increase in sEMG amplitude indicates that the exoskeleton robot we used is lightweight, provides good man–machine compatibility, and the QDD actuator resistance is small, which can effectively ease the burden of the wearer and facilitate better motion control. By observing the sEMG amplitude curve of VM, it can be found that the peak value of VM is similar in the two states, but the peak duration is longer in the EXO-OFF state.

Through the torque-control model we designed and the different values of the time-shift parameter, the sEMG signal amplitude of the RF and the VM decreased to different degrees when the knee joint was climbing up stairs compared with the NO-EXO state. When $t_{s}=0$, the sEMG signal amplitude of the RF decreased by 14.64% relative to the NO-EXO state, the sEMG signal amplitude of the VM decreased by 14%, and the sEMG signal amplitude of both muscles decreased by 14.36% overall. When $t_{s}=20\;ms$, the sEMG signal amplitude of the RF decreased by 17.93% relative to the NO-EXO state, the sEMG signal amplitude of the VM decreased by 16.5%, and the sEMG signal amplitude of both muscles decreased by 17.31% overall. When $t_{s}=50\;ms$, the sEMG signal amplitude of the RF decreased by 20.87% relative to the NO-EXO state, the sEMG signal amplitude of the VM decreased by 17.45%, and the sEMG signal amplitude of both muscles decreased by 19.37% overall. By observing the curve in Figure 13, it can be seen that when the exoskeleton robot starts to work, the peak sEMG signals of both the RF and the VM muscles decrease relative to the NO-EXO state. As $t_{s}$ gradually approaches 50 ms, the decrease in the amplitude of the sEMG signals of the two muscles relative to the NO-EXO state gradually increases. When $t_{s}=20\;ms$, the decrease in the amplitude of the RF and the VM sEMG signals was 3.29% and 2.5% greater than when $t_{s}=0$, and the overall decrease was 2.95% greater. When $t_{s}=50\;ms$, the decrease in the amplitude of the RF and the VM sEMG signals was 2.94% and 0.95% greater than when $t_{s}=20\;ms$, and the overall decrease was 2.06% greater. SEMG experiments verify the effectiveness of our torque-control model and parameter-optimization method.

The current research study has some limitations. Firstly, regarding model generalization, the current experiment is limited to five healthy young participants and is limited to assisting during climbing up stairs. In the future, experiments should include more participants to validate the applicability of this method in a broader context. We will continue to research control models for more complex assistance needs, such as walking down stairs and walking on slopes. Secondly, there is a certain degree of jitter in the output results of the current regression-forecasting model, as shown in Figure 11. In future research studies, filtering may be added to the existing model output results to make them smoother. This may introduce new delays and require recalculating the optimal time shift parameters and further experimental verification.

## 5. Conclusions

This article proposes a torque-regression forecasting model and parameter optimization method for knee exoskeleton robots. The regression-forecasting model combines CNN and LSTM networks, which have good data-feature-extraction and time-series processing abilities. The input of the model is the IMU data of the lower limbs, and the output is the torque of the knee joint. The data during the training stage come from an optical motion-capture system and three-dimensional force-measuring tables. To compensate for system delay, this research study proposes a method of time shifting the torque curve during the model training stage. This research study used RMSE to evaluate the difference between the ideal curve and the actual curve to obtain the optimal time shift parameter $t_{s}=50\;ms$. The experimental results of sEMG signals indicate that the torque-regression-forecasting model in this research study can effectively reduce the amplitude of the RF and the VM sEMG signals. As the experimental parameters approach the optimal value, the amplitude of the sEMG signal of the tested muscle gradually decreases. The effectiveness of the torque-control model and parameter-optimization method proposed in this research study was verified through sEMG signal experiments. With the assistance of the above model, the sEMG activity of the RF decreased by 20.87%, while the sEMG activity of the VM increased by 17.45%.

## Figures and Tables

**Figure 1 sensors-24-01505-f001:**
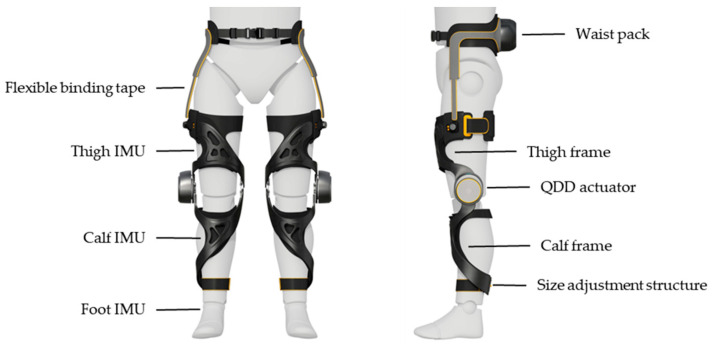
The proposed knee exoskeleton robot, including a waist pack, a set of bindings, two thigh frames, two QDD actuators, two calf frames, and six IMUs.

**Figure 2 sensors-24-01505-f002:**
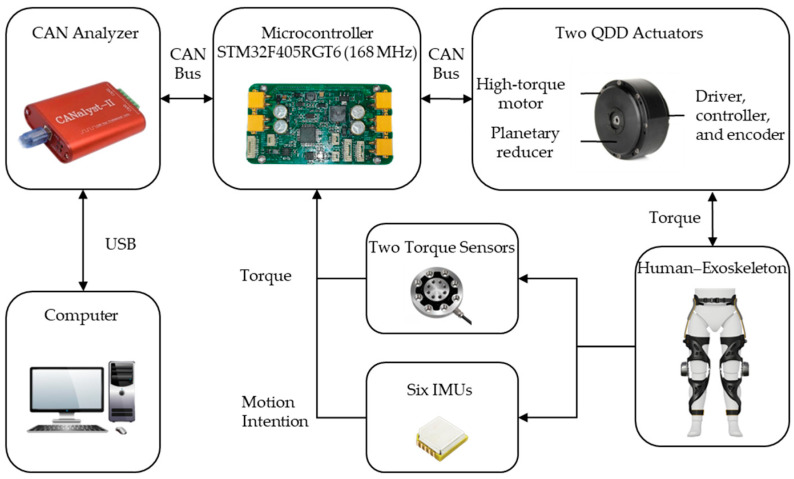
Electronic hardware architecture and QDD actuator.

**Figure 3 sensors-24-01505-f003:**
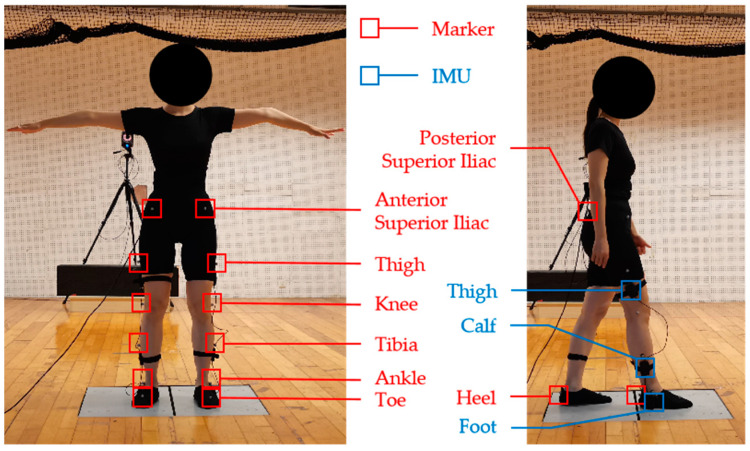
The schematic diagram of the reflective markers and IMUs.

**Figure 4 sensors-24-01505-f004:**
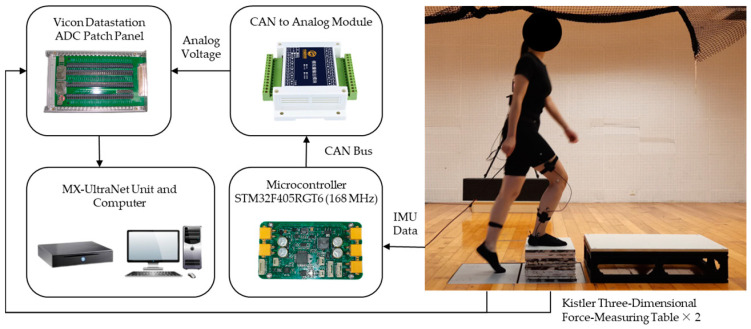
The block diagram of data-acquisition system.

**Figure 5 sensors-24-01505-f005:**
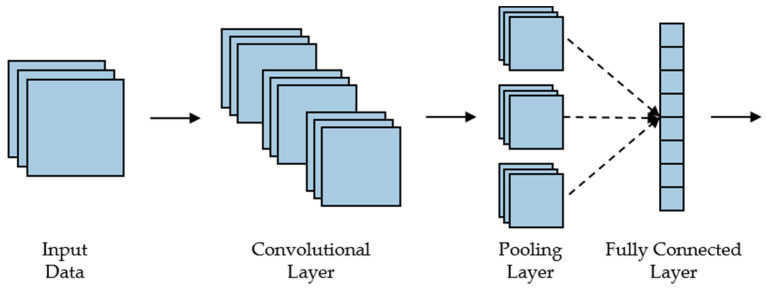
The main structure of CNN.

**Figure 6 sensors-24-01505-f006:**
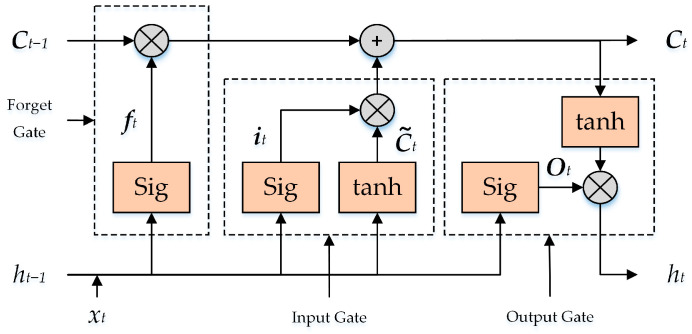
The LSTM structure diagram.

**Figure 7 sensors-24-01505-f007:**
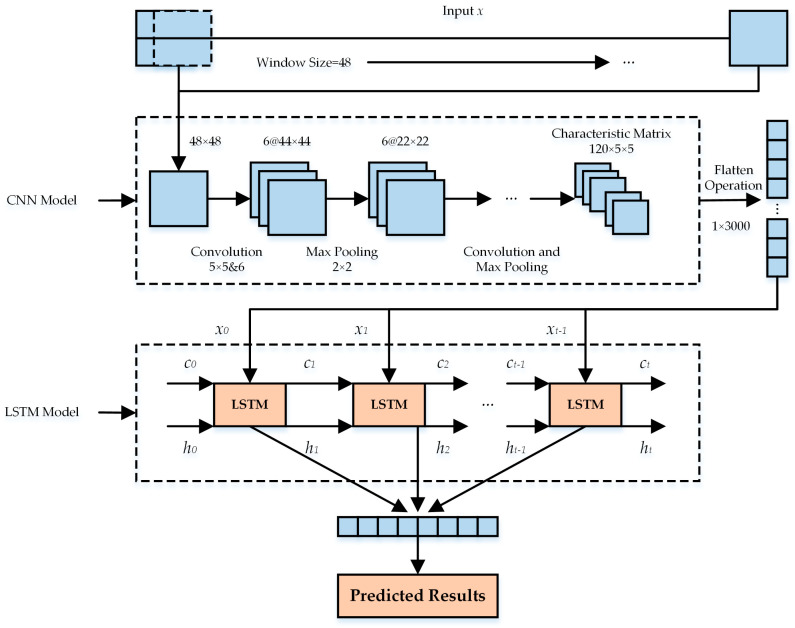
The structure of the combined CNN-LSTM model.

**Figure 8 sensors-24-01505-f008:**
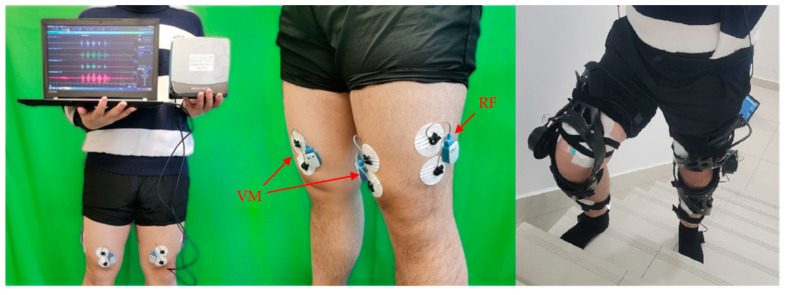
The locations of the sEMG sensors.

**Figure 9 sensors-24-01505-f009:**
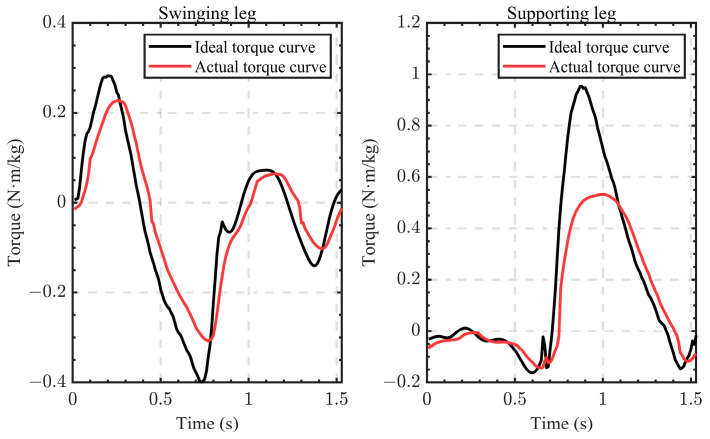
A set of ideal torque curves of the knee joint and the actual torque curves of the motor actuator acting on the joint.

**Figure 10 sensors-24-01505-f010:**
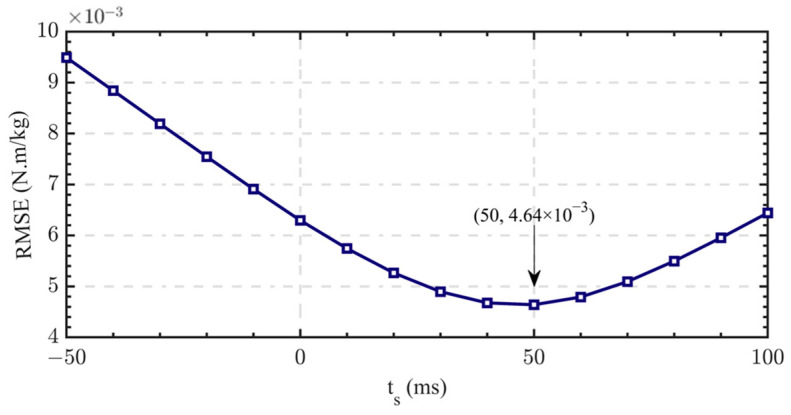
The change curve of weighted average accuracy with time shift parameter.

**Figure 11 sensors-24-01505-f011:**
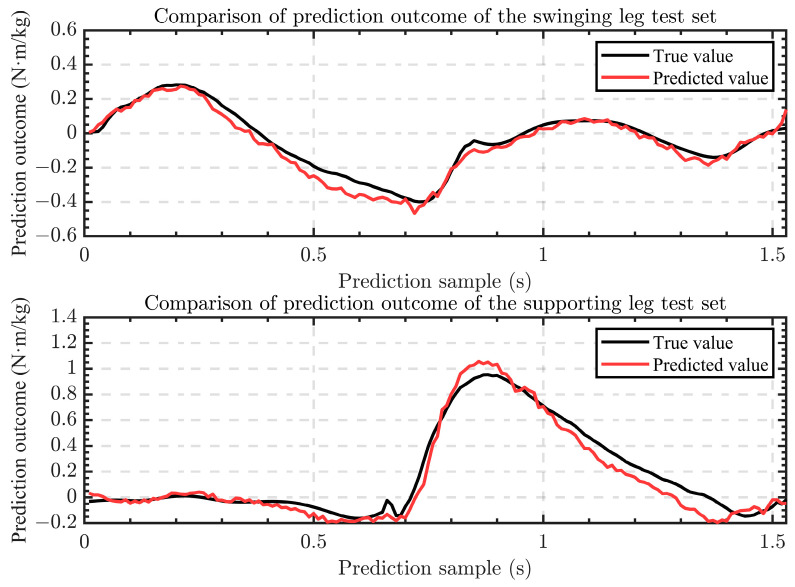
The prediction results when $t_{s}=0$.

**Figure 12 sensors-24-01505-f012:**
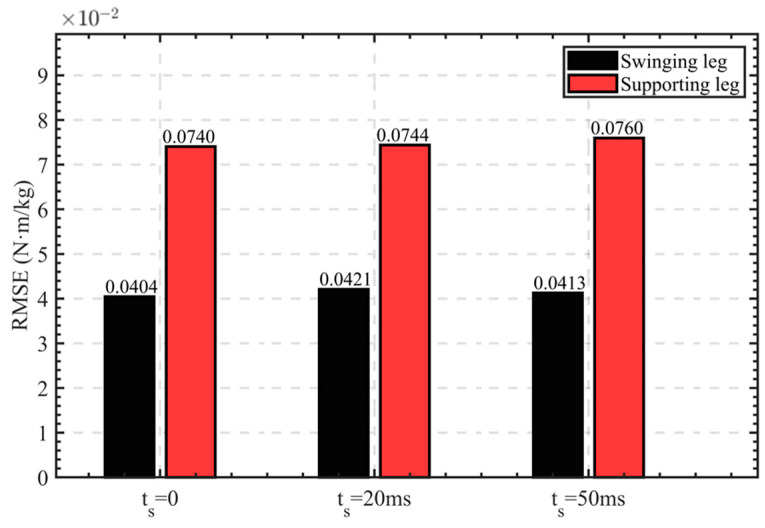
The RMSE under each parameter.

**Figure 13 sensors-24-01505-f013:**
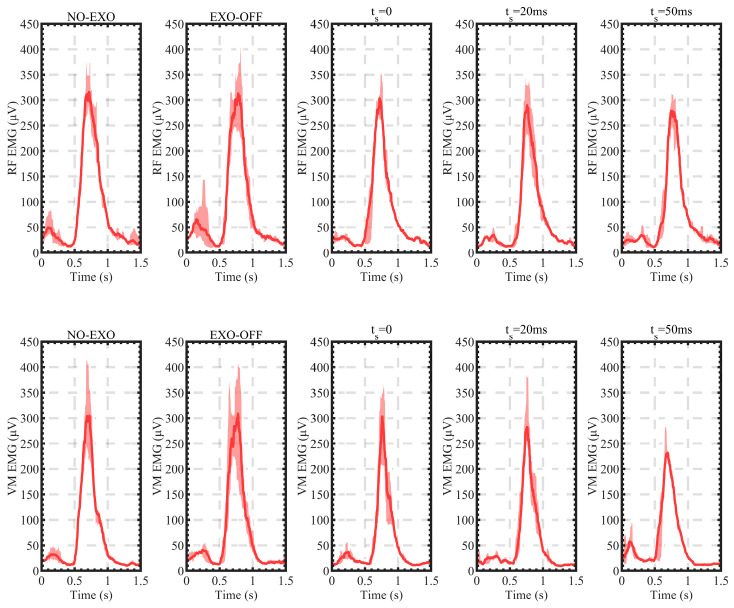
Mean RMS values of the sEMG signal amplitudes under various assisting situations.

**Table 1 sensors-24-01505-t001:** The details of the subjects.

NO	Height (cm)	Weight (kg)	Age	Gender
1	175	56.5	27	male
2	173	84	26	male
3	162	55	27	female
4	173	86.6	27	male
5	170	85.4	25	male

**Table 2 sensors-24-01505-t002:** Max RMS value under different assistance functions.

Serial	EXO-OFF	$t_{s}=0$	$t_{s}=20\;\mathrm{ms}$	$t_{s}=50\;\mathrm{ms}$
RF	−0.12%	14.64%	17.93%	20.87%
VM	−8.57%	14%	16.5%	17.45%
Overall	−3.81%	14.36%	17.31%	19.37%

## Data Availability

The data are not publicly unavailable due to ethical and privacy issues.

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
