# Peer review of "Optimization of Torque-Control Model for Quasi-Direct-Drive Knee Exoskeleton Robots Based on Regression Forecasting"

_sensors, 2024, doi:10.3390/s24051505_

Round 1

Reviewer 1 Report

Comments and Suggestions for Authors

The paper presents the optimization of torque control model for knee exoskeleton robots based on regression forecasting. The robot is actuated by quasi-direct-drive actuators. The authors should consider the following comments

1. The manuscript lacks an ethics statement, which is crucial considering the human involvement in the experiment. Please ensure the inclusion of an ethics statement in the paper.

2. The CNN-LSTM model employs matrix X as input and produces knee torque (ht) as output. It is recommended that the authors provide theoretical equations or reference established theories to elucidate how knee torque is exclusively dependent on the components of X. This will enhance the clarity and understanding of the model.

3. The units of all parameters need to be revised to adhere to the SI unit system. For instance, the unit of torque is currently expressed as N.mm/kg, which may not be standard. Additionally, ensure that the units of the Root Mean Square Error (RMSE) in the figures are consistent and clearly indicated.

4. In the equations, refrain from using the dot as a multiplication symbol. Consider using '\times' or simply remove the dot for improved clarity and adherence to standard mathematical notation.

Author Response

Dear Reviewer,

Firstly, I would like to express my deepest gratitude to you for taking the time out of your busy schedule to review our paper and provide valuable suggestions and feedback. We are truly appreciative of your efforts.

For the your comments and suggestions, we make point-by-point replies as follows.

1. In the new manuscript, we have added the ethical statement in line 246. Our experiment has been approved by the Ethics Committee of Soochow University, numbered SUDA20221228H08. The relevant materials have been emailed to the editor.

2. We have supplemented some existing research results to reflect the correlation between lower limb kinematics data collected by IMU and lower limb joint torque. The details can be found in lines 311 to 327 of the manuscript, and related research can be found in references [29] to [31].

3. We have checked and modified the units of parameters. Specifically, we changed the torque unit from N.mm/kg to N.m/kg, and the time unit from ms to s in Figure 9 and Figure 11. We changed the time unit from ms to s in Figure 13. The value of some data is relatively small, such as the time shift parameter and the amplitude of the EMG signal, we still retain ms and uV as units, hoping to be understood. We have indicated the units of the RMSE in Figure 10 and Figure 12.

4. We have used ‘\times’instead of dot as the multiplication symbol. For details, see formula (1) to Formula (6).

Overall, we have taken your review comments very seriously and made our best efforts to make the necessary modifications and improvements. We hope that our paper now better addresses your previous concerns and is ready for further review.

Thank you once again for your valuable suggestions and assistance. We hope that our paper has the opportunity to gain your approval.

Reviewer 2 Report

Comments and Suggestions for Authors

In this paper, a torque regression prediction model and a parametric optimization knee exoskeleton robot are proposed, which combines convolutional neural network and LSTM network with regression prediction model, the input of the model is the IMU data of the lower limb and the output is the torque of the knee joint. The data from the training phase comes from the opti-CAL motion capture system and the 3D force plate. The use of RMSE to assess the difference between the ideal curve and the actual curve has a certain depth of research, and the references listed in the last five years are appropriate. 

Comments on the Quality of English Language

There are certain aspects that need to be improved, such as that all units (e.g. ms) should be Times New Roman and orthodox; Functions (e.g. tanh) should be orthographic.

Author Response

Dear Reviewer,

Firstly, I would like to express my deepest gratitude to you for taking the time out of your busy schedule to review our paper and provide valuable suggestions and feedback. We are truly appreciative of your efforts.

For the your comments and suggestions, we make point-by-point replies as follows.

1. We have checked and modified the font and format of all the units. According to the template of this journal, we have modified the fonts of all units and formulas to Palatino Linotype, hoping to be understood. We modified the units of some parameters to adhere to the SI unit system. Specifically, we changed the torque unit from N.mm/kg to N.m/kg, and the time unit from ms to s in Figure 9 and Figure 11. We changed the time unit from ms to s in Figure 13. We have indicated the units of the RMSE in Figure 10 and Figure 12.

2. We have checked and modified the format of all the functions. For some special functions, such as Sig and tanh, we explain their meaning in line 277 of the manuscript.

Overall, we have taken your review comments very seriously and made our best efforts to make the necessary modifications and improvements. We hope that our paper now better addresses your previous concerns and is ready for further review.

Thank you once again for your valuable suggestions and assistance. We hope that our paper has the opportunity to gain your approval.

Round 2

Reviewer 2 Report

Comments and Suggestions for Authors

The author changed all the symbols to bold this time and mistakenly changed the variables to bold as well. (1) Need to change variables to italics; (2) Reference numbers such as [1] need to be superscripted.

Author Response

Dear Reviewer,

Firstly, I would like to express my deepest gratitude to you for taking the time out of your busy schedule to review our paper and provide valuable suggestions and feedback. We are truly appreciative of your efforts.

For the your comments and suggestions, we make point-by-point replies as follows.

  1. We have carefully checked all variable formats and have modified them to italic.
  2. We have carefully consulted the Microsoft Word template of this journal as well as other articles already published in this journal. The link to the template is 'https://www.mdpi.com/files/word-templates/sensors-template.dot'. We noticed that they did not use superscript for reference numbers. We hope for your understanding.

Overall, we have taken your review comments very seriously and made our best efforts to make the necessary modifications and improvements. We hope that our paper now better addresses your previous concerns and is ready for further review.

Thank you once again for your valuable suggestions and assistance. We hope that our paper has the opportunity to gain your approval.